# MACHINE LEARNING PIPELINES SYNTHESIS WITH LARGE LANGUAGE MODELS

## ABSTRACT

In the realm of machine learning, the ability to seamlessly translate natural language descriptions into compilable code is a longstanding challenge. This paper presents a novel framework that addresses this challenge by introducing a pipeline capable of iteratively transforming natural language task descriptions into code through high-level machine learning instructions. Central to this framework is the fine-tuning of the Llama model, enabling it to rank different solutions for various problems and select an appropriate fit for a given task. The paper covers the fine-tuning process and provides insights into the general process of transforming natural language descriptions into code. Our approach marks a significant step towards automating code generation, bridging the gap between task descriptions and executable code, and holds promise for advancing machine learning applications across diverse domains. We showcase the effectiveness of our framework through experimental evaluations and discuss its potential applications in various domains, highlighting its implications for advancing the field of machine learning.

## 1 INTRODUCTION

In recent years, there have been significant advancements in the field of code generation and text-to-code conversion, primarily propelled by the advent of Large Language Models (LLMs). Noteworthy contributions from leading researchers such as Codex (Chen et al., 2021), AlphaCode (Li et al., 2022), PaLM-Coder (Chowdhery et al., 2022), GPT models (OpenAI, 2023) have played a significant role in these advancements.

Alongside these works, a wave of exploration efforts has emerged, aiming to enhance the alignment between end-user expectations and the output generated by LLMs. These endeavors range from "chain of thoughts" to zero-shot learning and even employing reinforcement learning techniques (as demonstrated by Shen et al. (2023)).

While code snippet generation has proved to have a solid base of scientific investigation, converting textual descriptions of a complex machine learning (ML) task into primarily suitable compilable code is a relatively developing area of research.

**Our Goal: Transforming ML Task Descriptions into Code.** In this paper, we take on the challenge of translating ML task descriptions, written in everyday language, into code. We've broken this task into two parts: first, we create high-level code instructions, and then we transform these instructions into Python code. This approach provides the flexibility to choose any programming language for implementation.

**Generating Instructions.** To extract the high-level code instructions, we've devised a four-stage framework, depicted in Figure 1. Here's how it works:

1. *High-Level Solution Representation*: We begin by creating high-level representations of ML solutions. Leveraging the capabilities of GPT-3, we extract critical information regarding data preprocessing, model architecture, and the training procedure from existing solutions.

2. *Llama Fine-Tuning*: We then employ Llama for fine-tuning, utilizing the distilled solution information as a response. This fine-tuning process incorporates the solution and the corresponding task description, task metadata, data specifics, and the evaluation metric type as prompt.

3. *Llama Inference*: The top three high-level instructions are obtained from Llama.

4. *Enhancing LLM Responses with Smart-GPT*: The inferenced instructions are refined using the smart-GPT technique. This approach enhances the LLM response through additional prompts (for more details, see Section 4).

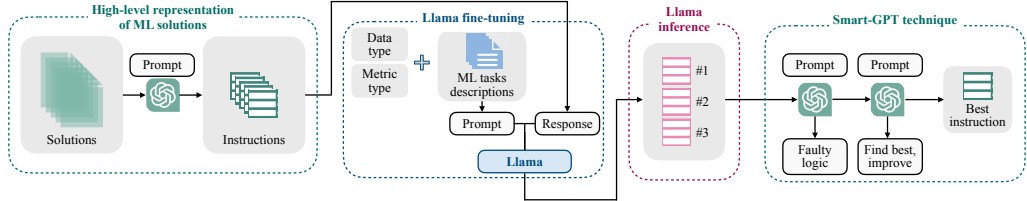

Figure 1: Machine learning task description to code high-level instruction transformation framework.

**Instruction-to-Code transformation.** The refined high-level ML instructions undergo sequential processing with GPT-4, resulting in the generation of functional code. One can find detailed insights into this instruction-to-code transformation process in Section 4.

**Empirical Validation in Machine Learning.** Our experimental results substantiate the effectiveness of our framework, particularly within the field of machine learning. We showcase that our approach can effectively produce code with promising results, as demonstrated by the evaluation metrics.

**Main Contributions.** To summarize, our main contributions are twofold:

1. *A Controllable Transformation Framework*: We introduce a framework for the controlled transformation of ML task natural language descriptions into suitable high-level solution instructions. The framework involves fine-tuning the Llama-2 model using pairs of ML task descriptions and instructions retrieved with GPT-3.

2. *Instruction-Based Sequential Generation*: We demonstrate that executing instructions for sequential generation leads to producing compilable code backed by promising results based on evaluation metrics.

In the subsequent sections, we delve deeper into each facet of our framework, providing empirical evidence of its effectiveness and highlighting its vast potential for diverse applications within machine learning.

## 2    RELATED WORK

The intersection of machine learning and code generation has garnered significant attention in recent years. We explore relevant research contributions, focusing on advancements in code generation, text-to-code conversion, and methods for aligning user expectations with model outputs.

## 2.1 CODE GENERATION AND TEXT-TO-CODE CONVERSION

Code generation has seen remarkable progress, attributed mainly to the rise of Large Language Models (LLMs). Feng et al. (2020) introduce a pre-trained model that can generate code and handle natural language tasks. Such models offer versatility in code-related tasks, including code generation, summarization, and recommendation.

CoditT5 (Zhang et al., 2022) is another language model that can generate the edit-based output sequence given the corrupted input sequence. Pre-trained models like CoditT5 are valuable resources for improving code generation capabilities and aligning them with user requirements.

Modern code generation approaches are based on general-purpose transformers such as GPT-3. Noteworthy among these models is Codex, which has showcased the potential to generate code snippets directly from natural language prompts. AlphaCode builds upon this foundation, emphasizing the importance of code diversity and improving the contextual understanding of LLMs.

In parallel, text-to-code conversion has gained prominence. PaLM-Coder presents a method for converting natural language descriptions into code, focusing on Java code generation. OpenAI (2023) and Bubeck et al. (2023) have further extended the capabilities of LLMs in understanding and generating code from textual prompts.

Controllable code generation is an emerging subfield with significant potential. Keskar et al. (2019) introduce conditional language models for controlled code generation. The authors focus on allowing users to specify conditions that influence the generated code, providing a level of control over the output.

Zhu et al. (2018) present a benchmarking platform for evaluating text generation models, including those designed for code generation. This platform facilitates the assessment of controllable code generation models by offering standardized evaluation metrics and tasks.

## 2.2 ALIGNMENT OF USER EXPECTATIONS

Efforts to align user expectations with LLM-generated code outputs have emerged as a crucial area of research. The "Chain of Thoughts" method, as demonstrated in (Hebenstreit et al., 2023), (Wei et al., 2023), leverages user-provided intermediate steps to guide code generation, ensuring the generated code aligns with the user's intentions. It involves breaking down complex programming tasks into logical and coherent steps akin to a thought process. This approach encourages the model to produce more structured and contextually accurate code. Researchers have explored the use of "Chain of Thoughts" to enhance the reliability and understandability of code generated from natural language, which is crucial in applications like code completion and automated software development.

Zero-shot learning, explored in (Thirunavukarasu, 2023), extends the capability of LLMs to generate code in languages it hasn't been explicitly trained on. This capability is achieved through pre-training on a vast corpus of text, enabling the model to generalize knowledge and apply it to new tasks or languages. Zero-shot learning has significant implications for cross-lingual NLP, code-switching, and multilingual text generation, broadening the versatility of language models.

"Smart GPT" (Nair et al., 2023), (Moghaddam & Honey, 2023) refers to relatively new techniques and methodologies employed to enhance the performance and reliability of GPT-based models. These techniques typically involve fine-tuning the model on specific tasks, introducing additional prompts or context, and refining the output to ensure logical coherence and correctness. Smart GPT is essential in addressing the limitations of language models, such as generating code that aligns with user expectations and task requirements. It can involve iterative processes where the model refines its responses based on user feedback or additional context, making it more adaptable and context-aware.

Reinforcement learning techniques, as detailed in [(Shinn et al., 2023), (Shen et al., 2023), (Thakur et al., 2022)], have been employed to fine-tune LLMs for code generation, enhancing the coherence and correctness of generated code. These techniques involve training language models like GPT using reinforcement learning frameworks, allowing them to learn from feedback and iteratively improve their code generation skills. Reinforcement learning can help models generate code that not only compiles correctly but also aligns with desired behavior or functionality. This approach has promising applications in automating code generation for specific tasks and improving the code quality generated from natural language descriptions.

### 2.3 MACHINE LEARNING PIPELINES GENERATION

While generating code snippets has been extensively explored, transforming textual descriptions into comprehensive and consistent machine learning workflows presents an ongoing challenge. Despite the state-of-the-art capabilities of GPT models, assessing generated code and precisely aligning it with the intended task remain areas of improvement.

Our approach focuses on the controlled transformation of complex ML task descriptions into high-level solution instructions. Such an approach expands the capabilities of code generation models into new domains and problem spaces, bridging the gap between natural language task descriptions and executable code.

## 3 DATA

For our research, we rely on the Code4ML dataset (Drozdova et al., 2023), a comprehensive resource comprising Python code snippets, contest summaries, and data descriptions from Kaggle competitions. It amplifies the dataset's value by incorporating essential competition-related metadata, which includes crucial details like data types and scoring metrics. These elements collectively render the Code4ML dataset an invaluable asset for addressing many challenges within the domain of machine learning.

The authors introduce a novel knowledge taxonomy tree to augment the dataset's utility further. This taxonomy tree is an innovative organizational framework that systematically categorizes Jupyter Notebook code snippets into various groups. Structuring the dataset in this manner effectively reduces the dimensionality of the learning space, thereby enhancing the efficiency of our pipeline generation process. This taxonomy provides a structured roadmap for understanding and navigating the dataset, streamlining the process of data utilization.

As mentioned in the paper, it's important to note that not all code snippets presented in the dataset can be unequivocally classified into specific taxonomy types. Nonetheless, the Code4ML dataset, with its wealth of information, remains a pivotal resource that significantly empowers our research in machine learning and code generation.

To enhance its usability for various machine learning tasks and code generation experiments, the dataset authors have meticulously organized the metrics into 20 distinct categories. These categories can be further classified into metrics designed for minimization and those tailored for maximization, providing a structured framework for evaluating ML solutions. We use this information to rank the solutions.

In our research, we concentrate on competitions covering all metric categories except 'points,' 'significance,' and 'custom loss.' From these competitions, we select the top 75 solutions for in-depth analysis and the retrieval of high-level instructions. It's worth noting that some contests have fewer than 75 solutions available for selection.

As a result, our dataset comprises 396 natural language ML task descriptions paired with 7023 corresponding instructions, obtained through the assistance of GPT-3, extracted from Kaggle solutions. Figure 2 offers an insightful overview of the prevalent models featured in the selected solutions. This analysis highlights

the dominance of specific models like LightGBM and EfficientNet in handling particular data types, underscoring their effectiveness and adaptability.

This dataset serves as the cornerstone of our experiments and training processes, providing the necessary foundation for exploring the transformation of ML task descriptions into high-level solution instructions.

| data_type/ metric | AUC | categorization accuracy | percentage errors | MSE | F-score multiclass | Multiclass log loss | Log loss | MAE |
|---|---|---|---|---|---|---|---|---|
| Audio | lightgbm | - | - | - | resnet-34 | - | - | - |
| Image | densenet-161 | efficientnet | - | Sequential | efficientnetb7 | vgg16 | logistic regression | lightgbm |
| Tabular | lightgbm | RandomForest classifier | lightgbm | lightgbm | lightgbm | xgboost | lightgbm | lightgbm |
| Text | RandomForest classifier | logistic regression | AUC | Bagging regressor | bert | logistic regression | xgboost | - |
| Video | - | - | - | - | - | - | myresnext | - |
| Time series | - | RandomForest classifier | RandomForest Regressor | lightgbm | - | bert | logistic regression | - |

Figure 2: The most popular model choice among Kaggle solutions based on metric and data type.

## 4 APPROACH

Drawing inspiration from the pioneering organizational framework presented by Drozdova et al. (2023), which systematically categorizes Jupyter notebook code snippets into various taxonomy groups, we adopt a similar concept for the purpose of dimensionality reduction in our ML task description-to-code synthesis approach. However, our focus diverges from the classification of individual code snippets. Instead, we pivot towards harnessing high-level information that encapsulates the essence of a generalized ML solution.

This strategic shift allows us to streamline the process of generating code from natural language task descriptions while operating more abstractly. Rather than delving into the intricate classification of code fragments, we aim to extract overarching patterns and critical components of ML solutions. This approach simplifies the synthesis process and promotes a more efficient and adaptable method of transforming task descriptions into executable code.

As previously outlined, our approach dissects the challenge of converting ML task descriptions into executable code in two distinct phases. We do it by introducing instructions for the high-level ML solution representations.

## 4.1 TASK DESCRIPTIONS TO INSTRUCTIONS SYNTHESIS

In the first phase, we tackle synthesizing high-level ML solution instructions from the provided ML task descriptions. This step involves the transformation of descriptive and often unstructured natural language expressions into coherent and actionable high-level instructions through the framework, as shown in Figure 1.

We initiate this phase by utilizing GPT-3 to extract vital information about data preprocessing, model architecture, and the training process from the provided code. Figure 3 illustrates the precise input prompt presented to the model. Consequently, GPT-3 generates the high-level ML instructions.

Subsequently, we employ the obtained instructions as a completion for fine-tuning Llama. Additionally, we provide LLM with the task description, metric details, and data type information as prompts. Figure 4 visually depicts the prompt-completion pair, separated by the [/INST] token, used in this stage.

---

**Prompt for GPT-3**

"Get the main information about data preprocessing, model architecture and model training from the code. Code: **Kaggle code**.

Figure 3: GPT-3 prompt for ML instructions retrieving.

---

**Prompt for GPT-3**

 token [INST] Get the main information about data preprocessing, model architecture and model training for this problem. This solution has the 1 place in rating. Data type: **Data type**. Metric type is: **Metric type**. Problem: **Task description**. [/INST] **GPT-resulted instruction** 

Figure 4: Llama fine-tune for ML instruction generation input.

---

Llama models have been pre-trained on vast amounts of data. By fine-tuning, we leverage this extensive knowledge and adapt it to specific tasks, often achieving state-of-the-art results with less data and time. The fine-tuning details are summarised in Appendix A.

## 4.2 INSTRUCTIONS REFINEMENT WITH SMART-GPT

Next, we select the top three most valuable instructions by specifying their rank using a dedicated prompt, as shown in Figure 5. The Llama temperature has been set up to 0.7. These inferred instructions then undergo further refinement with the assistance of smart-GPT. The primary goal of smart-GPT is to identify any logical errors in the provided instructions and subsequently choose the best option from the three variants, thereby enhancing the overall quality of the instructions. This intelligent processing is elucidated in Figure 6.

## 4.3 INSTRUCTIONS TO CODE GENERATION

The second phase of our approach centers on the actual generation of code, building upon the high-level instructions obtained in the previous step. In this phase, we harness the capabilities of language models to transform these instructions into functional and well-structured code that aligns seamlessly with the underlying ML tasks.

Figure 7 provides a visual representation of the sequential pipeline involved in the instruction-to-code transformation. We have broken down the code synthesis into distinct stages for Data Preprocessing, Model Architecture, and Model Training. Additionally, we have introduced a submission block to enable the testing of results on the Kaggle platform. The final step in this pipeline involves the seamless integration of all the generated code segments.

This phase forms the critical bridge between the high-level ML instructions and the executable code, ensuring that the generated code not only adheres to the provided instructions but also produces practical solutions for the intended ML tasks.

**Prompt for fine-tuned Llama-2**

Imagine that you are a data analyst. Your objective is writing the **3rd** place instruction for solving this machine learning task. Task: **Task description**. The **Data type** data is used for the problem. The metric type is **Metric type** for the problem. Your response contains the main information about data preprocessing, model architecture and model training.

Figure 5: Prompt for Llama inference.

**Prompt 1 for GPT-4**

You are a researcher tasked with investigating the 3 options of instruction for solving this machine learning task. Task: **Task description**. The **Data type** data is used for the problem. The metric type is **Metric type** for the problem.

Your response contains the main information about data preprocessing, model architecture and model training. List the flaws and faulty logic of each instruction option. Let's work this out in a step by step way to be sure we have all the errors.

Instruction option 1: **Instruction 1**
Instruction option 2: **Instruction 2**
Instruction option 3: **Instruction 3**

**Prompt 2 for GPT-4**

You are resolver tasked with 1) find which of the instruction options the researcher thought was best 2) improving the instruction 3) printing the improved instruction in full. Let's work this out in a step by step way to be sure we have the right meaningful instruction.

Figure 6: Smart-GPT for best instruction choice and improvement.

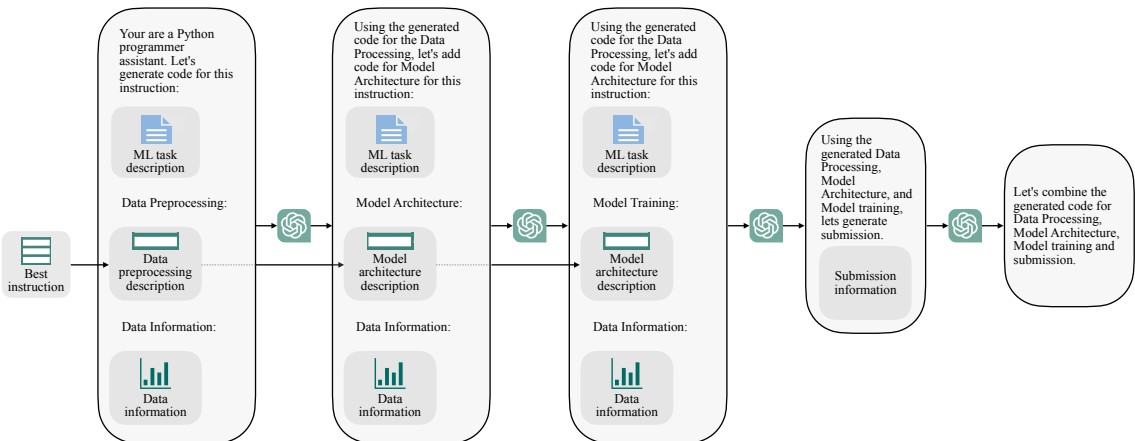

Figure 7: High-level instruction to code sequential transformation scheme.

These meticulously designed phases collectively form the foundation of our approach, enabling the seamless transition from unstructured ML task descriptions to precise, high-level solution instructions and, ultimately, to executable code.

## 5 RESULTS

To validate the effectiveness of our approach, we select Kaggle competitions that are not included in our training data. We apply our text-to-code pipeline by providing the natural descriptions of these chosen competitions along with the necessary meta-data to our framework. The summarized results for five sample ML tasks are presented in Table 1. The rating column corresponds to the position of our automatically

generated solutions out of the publicly available solutions number. It is evident that the generated code is not only compilable but also aligns well with the evaluation metrics.

Table 1: The sample results of generated ML code validated on Kaggle platform.

| Competition name | Metric | Score | Rating[1] |
|---|---|---|---|
| Feature Imputation with a Heat Flux Dataset[2] | rmse | 0.059 | 1/694 |
| Binary Classification of Machine Failures[3] | roc auc | 0.948 | 999/1502 |
| Predict CO2 Emissions in Rwanda[4] | rmse | 15.409 | 395/1440 |
| CommonLit - Evaluate Student Summaries[5] | rmse | 0.470 | 868/1903 |
| Binary Classification with a Tabular Credit Card Fraud Dataset[6] | roc auc | 0.773 | 446/641 |

It is noteworthy to consider the comparative efficacy of SmartGPT and unprocessed llama instructions in generating optimal solutions. While SmartGPT, a sophisticated technique, is adept at providing refined outputs, it occasionally exhibits a performance that is suboptimal compared to the unprocessed instructions derived from Llama. Comprehensive analysis reveals that the integration of three options from Llama does not consistently translate to enhanced solution quality. This inconsistency can be attributed to the inherent complexity and variability in the tasks, where the context and nuanced requirements play a pivotal role. Consequently, it underscores the necessity for a meticulous evaluation and potential refinement of the algorithmic approach employed by SmartGPT to ensure its consistent superiority in generating solutions of the highest quality.

It's worth noting that Kaggle, as a competitive platform, traditionally demands a significant investment of time and expertise from its participants. Engaging in Kaggle competitions often requires a deep understanding of the field and a substantial time commitment.
In contrast, our pipeline for transforming ML task descriptions into code offers a significantly more efficient alternative. This approach minimizes the time (it takes less than 1 minute to generate a solution) and expertise required to bridge the gap between task descriptions and executable code, making machine learning development more accessible and efficient.

## 6    CONCLUSION

In conclusion, our research presents a comprehensive approach for the transformation of unstructured ML task descriptions into executable code. Leveraging the Code4ML dataset, which encompasses a rich collection of Python code snippets, contest summaries, and data descriptions from Kaggle competitions, our methodology capitalizes on the dataset's valuable competition-related metadata, data types, and scoring metrics.

Drawing inspiration from the innovative knowledge taxonomy tree introduced by the dataset authors, we adopt a similar organizational framework to achieve dimensional reduction in our ML task description-to-code synthesis approach. However, our approach differs by focusing on high-level information extraction rather than individual code snippet classification. This strategic shift simplifies and streamlines the code generation process, making it more efficient and adaptable.

---

[1]The generated solution Kaggle links will be provided after double-blind review.

[2]Kaggle link: https://www.kaggle.com/competitions/playground-series-s3e15

[3]Kaggle link:https://www.kaggle.com/competitions/playground-series-s3e17

[4]Kaggle link:https://www.kaggle.com/competitions/playground-series-s3e20

[5]Kaggle link:https://www.kaggle.com/competitions/commonlit-evaluate-student-summaries

[6]Kaggle link:https://www.kaggle.com/competitions/playground-series-s3e4

Our approach is structured into two distinct phases: synthesizing high-level ML solution instructions and transforming these instructions into functional code. GPT-3 is employed to extract essential information from the provided code, which is the basis for generating high-level instructions. These instructions are then fine-tuned using Llama. The top three instructions are selected and further refined with the assistance of Smart-GPT, ensuring the highest quality instructions for subsequent code generation.

The second phase involves translating these refined instructions into well-structured and executable code segments, encompassing data preprocessing, model architecture, model training, and submission block generation. This transformation bridges the gap between high-level ML instructions and practical code, ensuring alignment with the underlying ML tasks.

Our approach's effectiveness is validated through experiments on Kaggle competitions that are not part of our training data. The results demonstrate that the generated code is compilable and aligns well with the specified evaluation metrics. We also compare the performance of Smart-GPT and unprocessed Llama instructions, highlighting the need for further refinement in Smart-GPT's algorithmic approach to achieve superior solution quality consistently.

In summary, our research provides an innovative and efficient solution for code generation from ML task descriptions. By capitalizing on the Code4ML dataset's wealth of resources and introducing a structured approach to instruction synthesis and code generation, we bridge the gap between natural language task descriptions and executable code, making machine learning development more accessible and efficient.

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

## A  LLAMA FINE-TUNING DETAILS

Table 2: Llama-2 fine-tuning hyper-parameters.

| LoRA Parameters | |
|---|---|
| LoRA attention dimension | 64 |
| Alpha parameter for LoRA scaling | 16 |
| Dropout probability for LoRA layers | 0.1 |
| **4-Bit Precision Parameters** | |
| Activate 4-bit precision base model loading | True |
| Compute dtype for 4-bit base models | float16 |
| Quantization type | nf4 |
| Activate nested quantization for 4-bit base models | False |
| **TrainingArguments Parameters** | |
| Number of training epochs | 1 |
| Enable fp16/bf16 training | False/False |
| Batch size per GPU for training | 4 |
| Batch size per GPU for evaluation | 4 |
| Number of update steps to accumulate the gradients for | 1 |
| Enable gradient checkpointing | True |
| Maximum gradient normal (gradient clipping) | 0.3 |
| Initial learning rate (AdamW optimizer) | 2e-4 |
| Weight decay | 0.001 |
| Optimizer | paged_adamw_32bit |
| Learning rate schedule | constant |
| Number of training steps | -1 |
| Ratio of steps for a linear warmup | 0.03 |
| Group sequences into batches with same length | True |
| Save checkpoint every X updates steps | 500 |
| Log every X updates steps | 25 |
| **Sequence Fine-Tuning Parameters** | |
| Maximum sequence length | None |
| Pack multiple short examples in the same input sequence | False |

