# OpenReview forum: "Machine learning pipelines synthesis with large language models"
_ICLR.cc/2024/Conference — ICLR 2024 Conference Withdrawn Submission_

### Official Review · Reviewer_561Y · 2023-10-18

**Soundness:** 1 poor
**Presentation:** 1 poor
**Contribution:** 1 poor
**Rating:** 1
**Confidence:** 5

**Summary:**

- This work aims to translate natural language descriptions of machine learning tasks into executable code
- Utilizing the GPT-3 model, this work first decomposes broad task descriptions into detailed sectioned instructions (instructions for data processing, model architecture definition, training, etc.) creating a dataset of task descriptions to detailed instructions.
- Next, it fine-tunes a Llama model to generate detailed instructions given a new task description
- Finally, it uses the detailed instructions generated by the Llama model as input to GPT-3 model to generate executable ML code
- This work introduces this pipeline and tests it on 5 Kaggle tasks and reports the evaluation metric score (rmse, and roc auc scores) along with where the solution would have ranked in comparison to other solutions.

**Strengths:**

- I found the decomposition of broad task descriptions into concrete subsection instructions interesting and it seems like a decent idea to explore.

**Weaknesses:**

- I don't think this work is anywhere close to publication. Honestly, it feels more like a project report submitted for publication, and it does not hold the rigor required for publication.
- There are some major issues with the paper, some of which I list below:
  - Related work: The entire related work section needs to be more thorough and relevant to the domain. Authors have cited works like the Sparks of AGI, Codex, and Palm papers, but do not mention numerous other works such as CodeT5, StarCoder etc.
    - Seemingly random papers were mentioned in the Related work, such as [1]. I could not figure out how this work explored zero-shot learning and how zero-shot learning extends the capabilities of LLMs to generate code in unseen languages!
    - The entire field of AutoML has not been mentioned, even though they do something similar to what this work aims to do.
- Approach: The entire paper is NL to ML code using GPT. Llama is used for prompt generation given natural language instructions, but the training data for it comes from the GPT model. Why can't the GPT model be used to decompose the description for the test data itself, especially when generating code from detailed instructions is done by the GPT model? What is the utility of the fine-tuned Llama model?
- Baselines: There are no baselines provided. How does the GPT model do without any prompt decomposition? How do other models perform?
- Test set size: Results are presented for 5 competitions. That is not enough sample size to judge the performance.
- Overall, the paper text does not do a good job of describing the approach well and it needs to be improved.


References:

[1] - Thirunavukarasu, Arun James. "Large language models will not replace healthcare professionals: curbing popular fears and hype." Journal of the Royal Society of Medicine (2023): 01410768231173123.

**Questions:**

None

---

### Official Review · Reviewer_Vjr6 · 2023-10-29

**Soundness:** 1 poor
**Presentation:** 1 poor
**Contribution:** 2 fair
**Rating:** 3
**Confidence:** 3

**Summary:**

This paper presents a framework that can address MLT tasks: first create high-level code instructions, then transform these instructions into Python code.  In their framework, there are four stages: High-Level Solution Representation, Llama Fine-Tuning, Llama Inference, Enhancing LLM Responses with Smart-GPT.

**Strengths:**

1. The topic is very interesting. The authors try to design a framework that can solve ML tasks with large language models. Several parts look good, for example, first create high-level code instructions, then transform these instructions into Python code.

2. In Table 1, the authors showed reasonable results in five competitions.

**Weaknesses:**

1. The presentation is bad. Several parts are unclear or missing. For example, what are the inputs for prompts in Figure 3 and Figure 4? What is "**Kaggle code**"? What is your training data from? In Section 4.1, the authors say that GPT-3 is used to generate high-level instructions. However, in Figure 1, it looks like OpenAI models are used. It is important that the authors have a clear description.


2. There are no detailed performance results: the performance of high-level code instruction creation or the quality of code generation. It is not enough to have sample results on 5 competitions ( shown in Table 1).

3. There is no baseline in this submission.

**Questions:**

Please see the above comments. A lot of parts of the submission are missing or unclear.  It is not ready for a complete conference submission.

---

### Official Review · Reviewer_sE3z · 2023-11-01

**Soundness:** 1 poor
**Presentation:** 2 fair
**Contribution:** 2 fair
**Rating:** 3
**Confidence:** 4

**Summary:**

Main Contribution:
- The paper introduces a novel framework for transforming natural language descriptions of machine learning tasks into compilable code. This is done in two phases - first generating high-level instructions for the ML solution using fine-tuned LLMs like Llama, and then transforming those instructions into executable Python code.

Novelty:
- Prior work has focused more on generating code snippets from prompts rather than full ML pipelines. This paper tackles the challenging problem of going from unstructured task descriptions to complete workflows.

- The approach of creating an intermediate representation of instructions and then using that to generate code is innovative. This provides flexibility and control over the final code.

Experiments:
- Experiments were done using the Code4ML dataset which has Python code, summaries and metadata from Kaggle competitions.

- 396 ML task descriptions were paired with 7023 instructions extracted from solutions using GPT-3. These form the training data.

- The framework was validated on unseen Kaggle competitions. The generated code compiled successfully and achieved competitive scores on the leaderboards.

- Comparisons were made between raw Llama instructions and refined SmartGPT instructions in terms of solution quality. The results were mixed, indicating that further refinements may be needed in SmartGPT.

Conclusion:
- The paper presents an end-to-end pipeline for going from natural language descriptions to executable ML code, using a combination of pre-trained LLMs.

- The instruction-based approach provides flexibility and control over the code generation process.

- Experiments demonstrate the framework can produce compilable, high-quality code for ML tasks. This makes ML development more efficient and accessible.

- There is scope for improving the solution refinement process using techniques like SmartGPT. Overall, the paper introduces a novel approach for controlled ML code generation from descriptions.

**Strengths:**

* Two-phase approach: The two-step approach of instructions followed by code generation provides flexibility and control lacking in other text-to-code techniques.

* Modular design: The overall pipeline is modular with clear interfaces between the stages.

**Weaknesses:**

1. Limited evaluation: The framework was only evaluated on a small set of unseen Kaggle competitions. More comprehensive evaluation on diverse ML tasks could strengthen the results.

2. No comparison to other text-to-code methods: The paper does not provide any quantitative comparison to other natural language to code generation techniques like PaLM-Coder, Code Llama, StarCoder etc. This could have demonstrated superiority.

3. Lack of ablation studies: Ablation studies could have isolated the contributions of different components like SmartGPT, inference with Llama, etc. This would give more insights.

4. No analysis of generated code quality: Beyond correctness, metrics analyzing code quality like modularity, comments, naming conventions could have provided more insights.

5. No discussion of limitations: The conclusion does not discuss any limitations of the current approach or challenges that need to be addressed in future work. Discussing limitations would have provided a balanced perspective.

Overall, while the paper introduces a novel framework, more rigorous evaluation and comparisons to other techniques could have strengthened the results. Providing more analyses of the generated code quality and limitations of the approach could have added valuable insights as well. Expanding the evaluation to more diverse tasks remains an area of future work.

**Questions:**

* This paper mentioned that this work is fine-tuning Llama. So does this paper mainly finetune Llama or Code Llama?

* Is it possible to compare the performance before and after fine-tuning?

---

### Official Review · Reviewer_qx6E · 2023-11-01

**Soundness:** 3 good
**Presentation:** 3 good
**Contribution:** 2 fair
**Rating:** 5
**Confidence:** 4

**Summary:**

This paper presents a framework for converting natural language machine learning task descriptions into code using large language models. The approach involves generating high-level instructions and transforming them into Python code. The framework demonstrates promising results in the machine learning domain, bridging the gap between task descriptions and executable code.

**Strengths:**

1. The paper presents a novel framework that addresses the challenge of transforming natural language machine learning task descriptions into executable code, leveraging large language models. This approach marks a significant contribution to the field of code generation and machine learning.
2. The paper demonstrates a well-designed framework, incorporating fine-tuning of the Llama model and instruction-based sequential generation. The experimental evaluations showcase the effectiveness of the approach, with promising results in the machine learning domain.
3. The paper is well-structured and easy to understand, providing a clear explanation of the proposed framework, its components, and the underlying methodology. The use of figures and tables effectively illustrates the approach and the results.
4. The proposed framework has the potential to advance machine learning applications across diverse domains by automating code generation and bridging the gap between task descriptions and executable code. This work holds promise for improving the efficiency and accessibility of machine learning development.

**Weaknesses:**

One potential weakness of the paper is the occasional suboptimal performance of SmartGPT compared to unprocessed Llama instructions. This inconsistency indicates the need for further refinement and evaluation of the algorithmic approach employed by SmartGPT to ensure consistently high-quality solutions.

Another limitation might be the generalizability of the framework. While the paper demonstrates promising results in the machine learning domain, it would be valuable to explore its applicability and effectiveness across a wider range of tasks and domains.

Lastly, the paper could benefit from a more comprehensive comparison with existing approaches in code generation and text-to-code conversion. Providing a deeper analysis of the framework's strengths and weaknesses in relation to other methods would help establish its position in the field and identify areas for further improvement.

**Questions:**

1. How does the proposed framework handle highly complex and large-scale machine learning tasks that may require specialized expertise? Can the framework adapt to the specific needs and intricacies of such tasks?
2. While the focus has been on Python code generation, how easily can the framework be extended to support other programming languages in the context of machine learning?
3. Have you considered any alternative approaches to refining the high-level ML instructions, apart from SmartGPT, to improve the quality and consistency of the generated solutions?
4. How does the framework account for the potential biases present in the training data, especially considering the reliance on the Code4ML dataset and Kaggle competition solutions?
5. Can you provide more details about the performance comparison between the proposed framework and other state-of-the-art methods in code generation and text-to-code conversion? This would help establish the advantages and limitations of your approach in relation to existing solutions.